# InstaInpaint: Instant 3D-Scene Inpainting with Masked Large Reconstruction Model

Junqi You[1,2]    Chieh Hubert Lin[1]    Weijie Lyu[1]    Zhengbo Zhang[3]    Ming-Hsuan Yang[1]

[1] UC Merced    [2] Shanghai Jiao Tong University    [3] Singapore University of Technology and Design

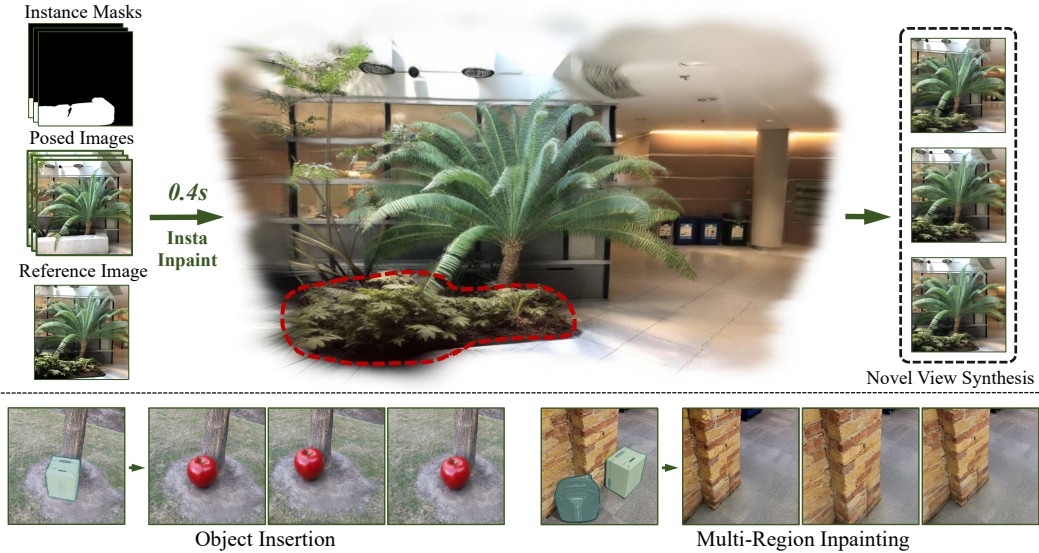

Figure 1: **Overview.** InstaInpaint generates an inpainted 3D scene from a set of posed images, a set of multi-view foreground instance masks, and a 2D reference image. The 3D reconstruction and inpainting process takes only 0.4 seconds. InstaInpaint is a generalizable framework supporting background inpainting, object insertion and multi-region inpainting simultaneously.

## Abstract

Recent advances in 3D scene reconstruction enable real-time viewing in virtual and augmented reality. To support interactive operations for better immersiveness, such as moving or editing objects, 3D scene inpainting methods are proposed to repair or complete the altered geometry. However, current approaches rely on lengthy and computationally intensive optimization, making them impractical for real-time or online applications. We propose InstaInpaint, a reference-based feed-forward framework that produces 3D-scene inpainting from a 2D inpainting proposal within 0.4 seconds. We develop a self-supervised masked-finetuning strategy to enable training of our custom large reconstruction model (LRM) on the large-scale dataset. Through extensive experiments, we analyze and identify several key designs that improve generalization, textural consistency, and geometric correctness. InstaInpaint achieves a $1000\times$ speed-up from prior methods while maintaining a state-of-the-art performance across two standard benchmarks. Moreover, we show that InstaInpaint generalizes well to flexible downstream applications such as object insertion and multi-region inpainting. More video results are available at our project page: https://dhmbb2.github.io/InstaInpaint_page/.

39th Conference on Neural Information Processing Systems (NeurIPS 2025).

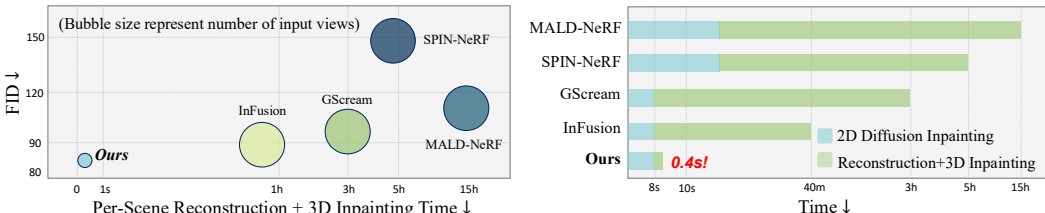

Figure 2: **3D Inpainting quality and speed.** Left: Our proposed method reconstructs the inpainted scene at a much faster speed with more competitive quality compared with existing approaches. Right: Our proposed method takes only 0.4s for reconstruction and 3D inpainting.

# 1    Introduction

Recent advances in neural reconstruction [1, 2, 3, 4, 5] achieve photorealistic and real-time rendering in virtual and augmented reality, enabling users to navigate within the digital twin of real-world environments freely. However, pure viewing without meaningful interactions with the digital content lacks practical applications, thereby motivating the rising interest in manipulating and editing the reconstructed 3D scenes. As existing frameworks all rely on scenes reconstructed with optimization-based methods before editing, it becomes intuitive to design editing algorithms that are also optimization-based. Unfortunately, such a design choice leads to lengthy operation time, causing inherently unbearable wait time and infeasible resource requirements. Some recent approaches mitigate the execution time problem with reference-based algorithm design, which first generates one 2D reference image and then achieves 3D edits by propagating the appearance to other views. Nevertheless, these methods still require impractical runtime, and regularizing the 3D geometry with a 2D appearance forms an ill-posed problem in which hand-crafted heuristics often fall short and lead to artifacts.

To address the execution speed problem, a natural solution is to leverage the Large Reconstruction Models (LRM) [6, 7] that can freeforwardly generate 3D geometry from sparse view images in less than a second. Learning from large-scale 3D scene data [8, 9], LRMs can produce high-quality and high-fidelity reconstructions even on unseen data. However, LRMs require input images to present consistent 3D information and solve the geometry using cross-view correspondence. In Fig. 3, we show that the cross-view 3D consistency generated from a state-of-the-art diffusion model (*i.e.* MVInpainter [10]) is insufficient for LRMs to solve a plausible geometry, leading to noticeable blurriness. In addition, multi-view diffusion models introduce substantial overhead in computational time, which conflicts with our real-time 3D inpainting goal. These observations motivate us to develop a single-stage and end-to-end method that learns to constitute 3D geometry within the LRMs.

We propose InstaInpaint, a new variant of LRMs that is tailored to achieve feed-forward reconstruction and edits simultaneously. Given a set of images paired with 3D-consistent 2D masks and one of the views being inpainted and served as the reference view, InstaInpaint predicts per-pixel Gaussian Splatting (GS) [7] parameters to reconstruct the scene. For pixels visible across views, the model still solves the geometry similarly to other LRMs. Meanwhile, the model learns to identify geometry from the surrounding context for pixels marked as reference pixels that have no geometric clues from other views. For instance, the extended geometry on the same plane should have a smooth depth transition, while inserted objects should have clear separation and stay in front of the background.

It is challenging to train such a model due to the lack of large-scale datasets that simultaneously provide (a) multi-view images with camera poses, (b) before-and-after image pairs where objects are physically removed, and (c) accurate masks of the objects being removed. We thereby design a self-supervised masked-finetuning scheme that utilizes a large-scale dataset that complies with (a) while circumventing the need for (b) and (c). In this work, we show that obtaining meaningful training masks is the most critical design. By masking the edited regions with gray pixels, we can compel the model to ignore the pre-edit appearance and directly produce post-edit results. We artificially create three types of masks: cross-view consistent object masks using an off-the-shelf video segmentation model [11], cross-view consistent geometric masks with LRM self-predicted depth, and randomly sampled image masks without cross-view consistency. In Section 4.2, we show that these masking mechanisms are essential to avoid object bias that leads to generalization failures. For each training sample, we subsample a few frames from a scene as InstaInpaint input views, while leaving the other frames as candidate supervision views. An input view is chosen as the reference view, while the editing regions in the remaining views are masked with gray pixels. InstaInpaint takes both masked input views, the reference view, and masks as input. Then, the network is end-to-end trained with

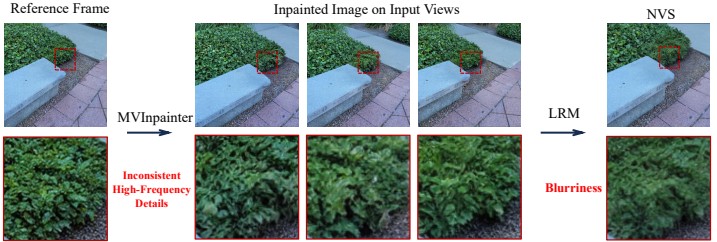

Figure 3: **Limitation of MVInpainter with LRM**. MVInpainter creates noticeable multi-view inconsistency in high-frequency regions, leading to blurriness in the final LRM reconstruction results.

photometric losses using the unmasked output views, serving as the post-edit ground-truth. As such, our design unlocks scalable training on large-scale real-world datasets.

We conduct extensive experiments on two standard 3D inpainting benchmarks with diverse and challenging real-world scenes. In Fig. 2, we highlight that InstaInpaint achieves state-of-the-art performance on both speed and quality axes. We also provide ablations of the key design choices on the masking strategies and the encoding design. The contributions of this work are:

- We propose InstaInpaint, a reference-based feed-forward framework that infers $1000\times$ faster than previous methods while achieving state-of-the-art quality across two 3D inpainting benchmarks.
- Our self-supervised masked-finetuning strategy unlocks efficient training on large-scale data.
- We demonstrate effective mask-creation strategies that avoid bias and improve generalization.
- InstaInpaint is a unified and flexible framework that simultaneously supports object removal, object insertion and multi-region inpainting without additional training.

## 2 Related Work

**Large Reconstruction Model.** Transformer-based 3D large reconstruction models (LRMs) [6] starts to emerge in recent years. With the scalable and generalization architecture and large 3d training dataset, LRMs are able to regress the 3D shapes from multi-view images in a feed-forward manner, represented as triplane NeRFs [6], 3D Gaussians [7, 12, 13, 14], latent tokens [15], or deformable Gassians [16]. Although LRM demonstrates high-quality sparse-view reconstruction ability, it struggles with inconsistent or even incomplete input views, limiting its application in 3D editing tasks. In this work, we aim to impart LRM the capability of deducing the texture and geometry of a multi-view inconsistent region from a single reference view. MaskedLRM [17] is a concurrent work that employs a similar masked training strategy for the mesh inpainting task in an object-centric setup using large-scale synthetic data. In contrast, InstaInpaint targets more challenging real-world scenes with complicated textures and geometries. Furthermore, our proposed self-supervised framework avoids the need for synthetic training data, avoiding the additional simulation-to-real domain gap.

**3D Scene Inpainting.** Existing 3D scene inpainting methods [18, 19, 20, 21, 22, 23, 24, 25, 26] mainly leverage the 2D image inpainting models to provide visual prior. Although some works [27, 28, 29, 10] try to enhance the 3D consistency of multi-view 2D inpainted results, there is still a notable textural shift in high-frequency details. One line of work paints each individual 2D image and addresses the 3D inconsistency problem with tailored designs. For instance, SPIn-NeRF [30] proposes replacing pixel loss with perceptual loss for better high-frequency detail reconstruction. InpaintNeRF360 [31] also develops similar strategies with perceptual and depth losses. MALD-NeRF [32] presents masked adversarial training and per-scene diffusion model customization for better consistency. While the carefully designed mechanics effectively enhance inpainting quality, inconsistency brought about by multi-view inpainting is hard to completely erase. Other works bypass multi-view 2D inpainting and adopt a reference-based inpainting strategy that imposes consistency by adhering to the provided reference image. This paradigm is adopted by some NeRF inpainting work [33], but is most common in 3DGS inpainting [34, 35, 36, 37, 38, 26]. For example, GScream [35] proposes a feature propagation mechanism to endow Gaussian blobs in inpainted areas with cross-view consistency. Infusion [39] starts with a partial Gaussian scene and directly optimizes the depth-initialized Gaussian blobs on the reference view. However, inpainted regions often exhibit a clear boundary, and foreground floaters tend to appear. In this work, we propose a reference-based feedforward model that guarantees the consistency and smooth integration of inpainted regions with self-supervised masked training.

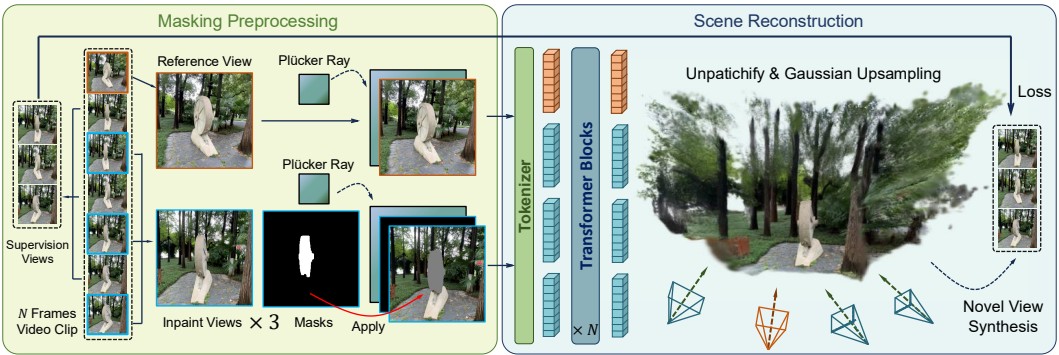

Figure 4: **Overall pipeline of Masked Finetuning.** Given a video clip, a reference view and three inpaint views are selected. The reference view remains intact - its RGB values and corresponding Plücker ray coordinates are directly tokenized. For the inpainting views, we first apply multi-view masks to the images, then concatenate these masked images with their Plücker coordinates and the binary masks before tokenization. Tokens are sent into transformer blocks to predict pixel-aligned Gaussians. Supervision views are randomly sampled from the remaining frames of the video clips to compute photometric loss against novel view renders.

## 3 Methodology

InstaInpaint is trained in two stages. We first train a naive LRM to equip the model with sufficient 3D reconstruction knowledge, then apply masked fine-tuning to perform the inpainting task. In Section 3.1, we give an overview of the model architecture and stage one training pipeline. We elaborate on the designs of the masked fine-tuning pipeline in Section 3.2 and the generation process of three types of masks in Section 3.3.

### 3.1 Overview

Large reconstruction models learn to regress 3D geometry directly from multi-view 2D images. Inspired by GS-LRM [7], InstaInpaint uses a ViT-based [40] self-attention transformer as its backbone. For a given set of multi-view input images $\{\mathbf{I}_i \in \mathbb{R}^{H \times W \times 3} \mid i = 1 \ldots N\}$, we first concatenate the images channel-wise, the Plücker ray coordinates [41] of each view $\mathbf{P}_i$. We patchify and linearize them into $H/p \times W/p$ tokens, where $p$ is the patch size. The patchifier is implemented as a $p \times p$ convolution kernel with stride $p$. These feature tokens $T_i$ are then flattened and concatenated as input to the self-attention transformer. Each transformer block consists of pre-Layernorm, multi-head self-attention, a MLP and residual connections. The output tokens $\mathbf{T}'_i$ from the transformer are then decoded into Gaussian parameters with a single linear layer:

$$\mathbf{T}_i = \text{Conv}_p(\text{Concat}(\mathbf{I}_i, \mathbf{P}_i)); \quad \mathbf{T}'_i = \text{Transformer}(\mathbf{T}_i); \quad \mathbf{G}_i = \text{Linear}(\mathbf{T_i}), \quad (1)$$

where $\mathbf{T_i}, \mathbf{T}'_i \in \mathbb{R}^{HW/p^2 \times D}$ and $D$ is the token dimension, $\mathbf{G}_i \in \mathbb{R}^{(HW/p^2) \times (p^2 q)}$ represents 3D Gaussians and $q$ is the number of Gaussian parameters. We then unpatchify $\mathbf{G}_i$ to a per-pixel Gaussian map $\mathbf{G}'_i \in \mathbb{R}^{H \times W \times q}$, where each 2D pixel has a corresponding 3D Gaussian. The final 3D scene is obtained by merging the multi-view per-pixel Gaussians, resulting in a total of $N \times H \times W$ Gaussians. We compute photometric losses on $M$ novel views. Following previous works [6, 7], we apply a combination of MSE loss and perceptual loss [42, 43].

### 3.2 Masked Finetuning Pipeline

In the second training stage, we employ a masked fine-tuning strategy to adapt LRM from 3D reconstruction to 3D inpainting using existing large-scale real-world datasets [8, 9].

**Selection of Input and Supervision Views.** As is shown in Fig. 2, for a given $N$ frames video clip and corresponding multi-view masks (we will detail the generation process in Section 3.3), we use its quartile frames as input frames (*e.g.*, the 1st, 5th, 10th, 15th frame for a 15 frames clip). We randomly pick a frame as the reference frame and keep the reference image complete. For the three non-reference input frames, deemed as inpaint views, we directly apply the multi-view masks onto the images by substituting the masked regions with gray pixels. We then randomly sample $M$ supervision views from the remaining $N - 4$ frames in the video clip and apply photometric losses. The masking process compels the transformer to predict cross-view consistent geometry for inpainted regions of the reference view that produce plausible content when projected onto novel views.

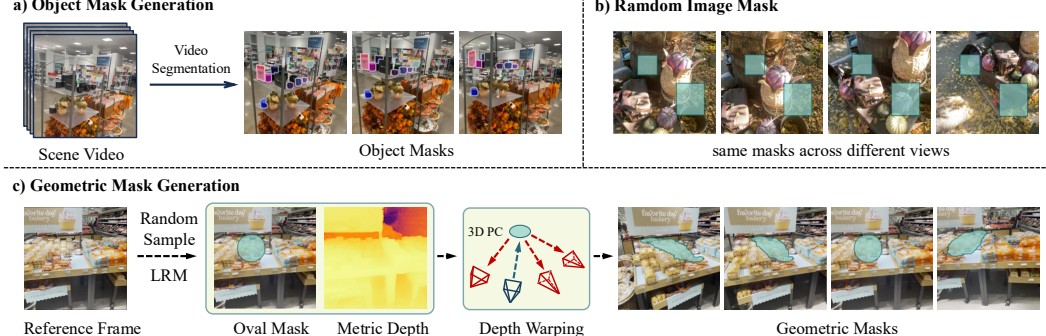

Figure 5: **Overview of mask generation methods**. **a)** Object Mask Generation. We propagate scene videos through a video segmentation model to obtain object masks. **b)** Random Image Mask. We randomly mask the same area for all input views. **c)** Geometric Mask Generation. We use a GS-LRM to predict metric depthmaps and then warp randomly sampled oval masks to inpaint views.

**Mask Encoding Strategy.** We directly encode multi-view masks to provide the model with additional information to recover geometry of missing regions. For inpaint views, the masked image is then channel-wise concatenated with binary multi-view masks before being tokenized.

We do not provide the inpaint region mask directly to the reference view. This means that when the reference image is fed into our pipeline, the binary mask image indicating the inpaint region is not given as an additional input alongside the reference RGB image itself, nor is the reference image's content explicitly altered (e.g., grayed out) in the masked region. This design is inspired by a key observation: the inpainted region and the reconstruction regions should maintain a similar data distribution during training. To clarify the model's perception of different areas, we conceptualize three kinds of tokens within our pipeline:

1. Unmasked regions with intact texture: These areas are fully visible and consistent across both reference and supervision views, providing clear multi-view stereo information vital for accurate 3D reconstruction.

2. "Masked regions" in the reference view with original texture: This refers to the conceptual area within the reference view that corresponds to the region proposed for inpainting. The model "sees" the original, unaltered texture in this region. However, since other non-reference views do have these corresponding regions explicitly masked out, the model is implicitly tasked with inferring the geometry and fine details of this area within the reference view based solely on its surrounding unmasked context. This unique setup is vital for maintaining a similar data distribution between token types (1) and (2), enabling the model to learn to faithfully reconstruct both areas equally well, whether the mask is explicitly presented or not.

3. Masked regions in non-reference views filled with gray pixels: These regions in non-reference views are explicitly altered (grayed out) with a provided mask and are provided with an explicit binary mask. This clear alteration forces the model to learn to ignore these areas or infer their content from other available views.

We aim for the model to faithfully reconstruct (1) using multi-view stereo vision, to accurately speculate the geometry of (2) based on neighboring regions in the reference view, and to mitigate negative effects from (3). This approach ensures a sufficient distinction between (2) and (3) to prevent feature contamination while promoting consistent learning across visible and conceptually "masked" areas.

$$\mathbf{T}_{\text{ref}} = \text{Conv}_p(\text{Concat}(\mathbf{I}_{\text{ref}}, \mathbf{P}_{\text{ref}})); \quad \mathbf{T}_{\text{nonref}} = \text{Conv}_p(\text{Concat}(\mathbf{I}_i \odot \mathbf{M}_i, \mathbf{P}_i, \mathbf{M}_i)) \ . \tag{2}$$

### 3.3 Mask Generation Process

Most current large-scale 3D datasets contain only videos captured in different scenes and camera annotations obtained from COLMAP [44]. In this section, we describe in detail how to obtain meaningful training masks from posed videos. As in Figure 5, we present three strategies for training masks that maximize data efficiency and model generalizability. At each iteration during training, we randomly pick one of these three types of masks based on an assigned probability.

**Object Mask**. We obtain multi-view consistent foreground instance masks by devising a data collection pipeline with an off-the-shelf video segmentation model [11]. For a given video clip, we first perform global segmentation on the first frame to acquire proposal prompts. We filter out background segmentation and shuttered artifacts by removing segmentation masks that are either excessively large (covering >50% of the image area) or excessively small (<0.5% of the image area). We also filter out masks too close to image edges, as they are likely to disappear in subsequent frames.

The segmentation masks collected $\{\mathbf{M}_i^0 \mid i = 1, \ldots, n\}$ are then transformed into axis-aligned bounding boxes to serve as visual prompts. The video segmentation model processes the video clip and produces a series of object masks $\{\mathbf{M}_i^t \mid i = 1, \ldots, n; t = 1, \ldots, T\}$ based on the generated bounding-box prompts. To prevent flickering artifacts that commonly appear in video segmentation, we perform a sanity check by filtering out masks whose Insection-over-Union value between two subsequent frames $(\mathbf{M}_i^t \cap \mathbf{M}_i^{t-1})/(\mathbf{M}_i^t \cup \mathbf{M}_i^{t-1})$ is too low. This is based on the heuristic that multi-view masks of two immediate frames should share a large common area. We cache the object masks for all frames in the clip. During training, we only use masks for selected input frames.

Although object masks provide high-quality multi-view consistent masking signals, object regions exhibit a distinct discontinuity from their neighboring regions, both in texture and in depth. However, during validation, we want the inpainted background to seamlessly integrate into the adjoining regions. This object bias leads to significant train-val gaps, causing the model to produce protruding depth in the inpainted area regardless of the actual geometry implied by the reference image. Therefore, we propose the following two types of masks to solve the problem.

**Geometric Mask**. Geometric masks cover a random multi-view consistent area. We collect geometric masks by warping masked pixels from reference views to inpaint views. Specifically, for the four selected unmasked frames $\mathbf{I}_i$ along with their camera poses and intrinsics $\mathbf{P}_i, \mathbf{K}_i$, we first send them into the GS-LRM trained in the first stage to predict per-view Gaussian maps $\mathbf{G}'_i \in \mathbb{R}^{H \times W \times q}$. We can extract the depthmap $\mathbf{D}_i \in \mathbb{R}^{H \times W}$ of each input view. For the reference frame, we first randomly sample a mask $\mathbf{M}_{\mathrm{ref}} \in \mathbb{R}^{H \times W}$ with one or more ovals in the pixel space. (Refer to Section A for more details on mask generation.) We project the masked pixels into 3D global coordinates, garnering a 3D point cloud:

$$\mathbf{X} := \left\{ \mathbf{P}_{\mathrm{ref}}^{-1} \, h\big( \mathbf{K}_{\mathrm{ref}}^{-1} \mathbf{D}_{\mathrm{ref}}(i,j)[i,j,1]^\top \big) \mid \mathbf{M}_{\mathrm{ref}}(i,j) = 1 \right\} \, , \tag{3}$$

where $\mathbf{P}_{\mathrm{ref}}$ is the camera extrinsic of reference view, $\mathbf{K}_{\mathrm{ref}}$ is the intrinsic for reference view, $\mathbf{D}_{\mathrm{ref}}(i,j)$ and $\mathbf{M}_{\mathrm{ref}}(i,j)$ means the depth value and mask value at pixel $[i,j]$ and $h : (x,y,z) \to (x,y,z,1)$ is the homogeneous mapping. The masking point cloud $\mathbf{X}$ is then projected onto the screen space of inpainting views, forming a set of pixel coordinates for each view. The inpainting masks are then created by assigning a value of true to all pixels within these projected regions.

$$\hat{\mathbf{X}}_i = \{\mathbf{K}_i \mathbf{P}_i \mathbf{x} \mid \mathbf{x} \in \mathbf{X}\}; \quad \mathbf{M}_i(p,q) = \Big[[p,q] \in \hat{\mathbf{X}}_i\Big]_{\mathrm{iver}} \, , \tag{4}$$

where $\mathbf{P}_i$ and $\mathbf{K}_i$ are the extrinsic and intrinsic for the $i$-th inpaint view, $[\cdot]_{\mathrm{iver}}$ means the Iverson bracket. We further apply morphological closing to the projected mask to reduce the shattered points. Note that we can directly warp the project point cloud from the reference frame because the depthmap predicted by GS-LRM is metric depth strictly aligned with input camera parameters.

**Random Image Mask**. Aside from the two types of cross-view consistent masks, we further improve the robustness of our model with cross-view inconsistent masks. Random image masks are generated by sampling four identical pixel-space masks, which contain one or more rectangular regions. The model learns to detect and reduce the inconsistency of the projected Gaussians during training.

## 4 Experiments

**Implementations.** InstaInpaint is trained on DL3DV [9]. which is one of the largest open-source real-world 3D multi-view datasets, featuring a diverse variety of scene types and camera motion patterns. Its training set, DL3DV-10k, contains 10k+ videos from both indoor and outdoor scenes. Each scene video contains 200~300 key frames with camera pose annotation obtained from COLMAP [44].

We rigorously follow the architectural designs of the transformer and the Gaussian upsampler from GS-LRM [7]. We set the image patch size to 8 and the token dimension to be 1024. In the first stage, we train the model on a resolution of $256 \times 256$ for 80K iterations. In the second stage, we finetune InstaInpaint on a resolution of $512 \times 512$. We add an additional channel to the image patchifier and initialize it with the mean of 3 original channels. We split each scene videos into 15-frame video

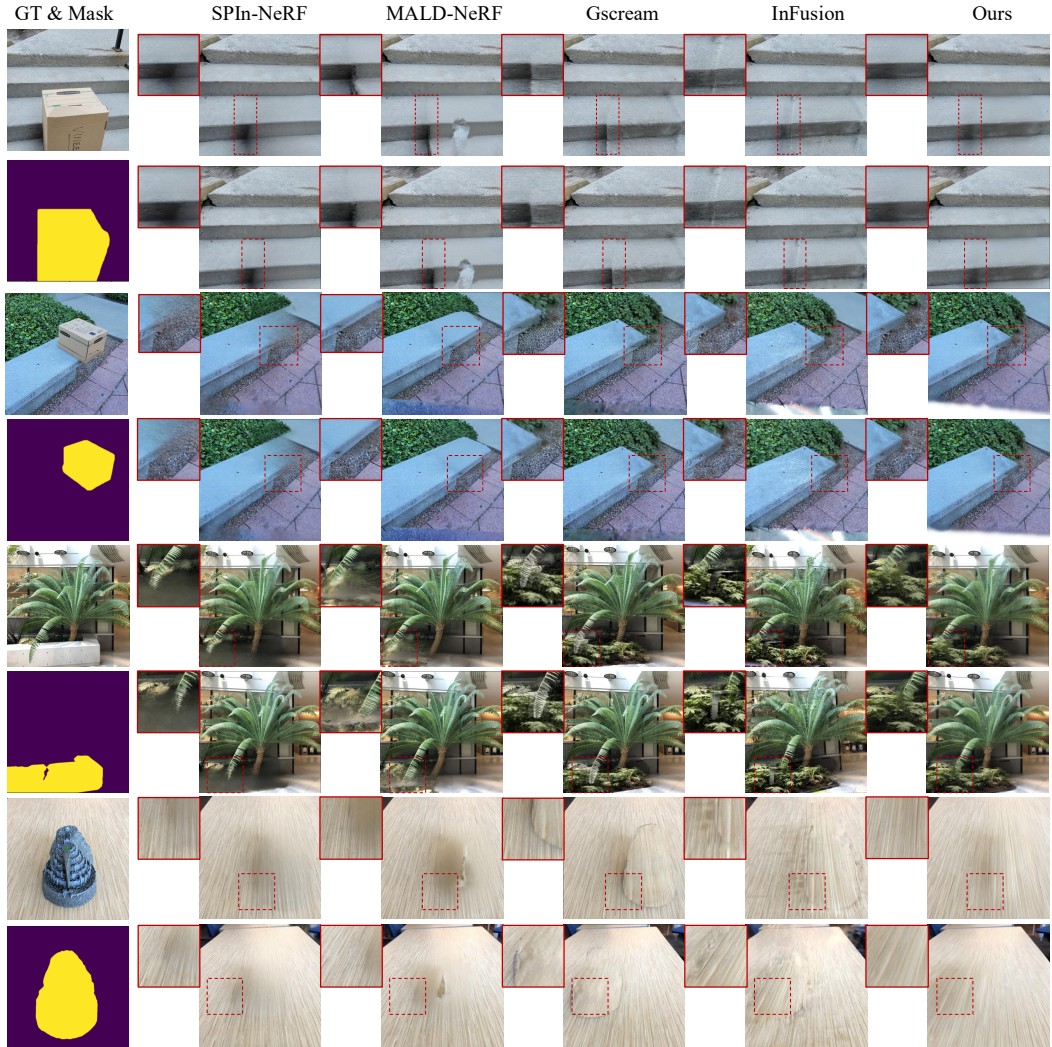

| GT & Mask | SPIn-NeRF | MALD-NeRF | Gscream | InFusion | Ours |

Figure 6: **Qualitative comparisons with state-of-the-art methods.**. InstaInpaint obtains more plausible inpainted texture, better consistency and smoother transition at the inpainting border. The top 4 rows are from SPIn-NeRF and the bottom 4 rows are from LLFF.

clips and randomly select one at each iteration. InstaInpaint takes 4 input views and is supervised on 8 views in both training stages. For all experiments, we train InstaInpaint for 12800 iterations in the second stage with 8 A6000 GPUs, which costs about 40 hours. We use Adam optimizer with a learning rate of $8e - 5$ and a total batch size of $80$. For training efficiency, we cache in advance all object masks and metric depthmaps predicted by LRM.

**Evaluation Datasets**. We evaluate InstaInpaint on two real-world datasets, SPIn-NeRF [30] and LLFF [45]. SPIn-NeRF is an object removal benchmark with 10 scenes. Each scene consists of 60 training views with foreground objects and 40 testing views with foreground objects removed. LLFF consists of several real-world 3D scenes with varying view numbers. Following prior work, we use a six-scene subset with SPIn-NeRF annotated 3D multi-view object masks. For InstaInpaint, We select the 4 input views from the training views set to ensure 1) the reference view is posed close to the center and 2) the three other context views cover an area as large as possible.

1. For reference view, we calculate the mean of all training views (60 for SPIn-NeRF) and select the view whose position is closest to the mean as the reference view. The reference view is inpainted with SOTA 2D-diffusion inpainting models [46].

2. For the three other views, we aim to choose three views that cover as wide range as possible. We first regress a plane from all training views. Then we project all training view positions onto the

Table 1: **Quantitative comparisons with state-of-the-art methods**. InstaInpaint not only achieves competitive results compared with previous optimization-based methods, but also demonstrates a great speed improvement. The best and second-best results are highlighted, respectively.

| Category | Method | Time↓ | SPIn-NeRF | | | | LLFF | |
| | | | LPIPS↓ | M-LPIPS↓ | FID↓ | KID↓ | C-FID↓ | C-KID↓ |
|---|---|---|---|---|---|---|---|---|
| None Reference-Based | SPIn-NeRF | 5h | 0.4973 | 0.3742 | 129.71 | 0.0256 | 228.86 | 0.0732 |
| | SPIn-NeRF+SD | 5h | 0.5274 | 0.4081 | 143.09 | 0.0295 | 235.36 | 0.0734 |
| | InpaintNeRF360 | 5h | 0.4562 | 0.3417 | 117.43 | 0.0216 | 229.85 | 0.0712 |
| | MALD-NeRF | 15h | 0.4240 | 0.3269 | 109.66 | 0.0192 | 223.71 | 0.0710 |
| Reference-Based | GScream | 3h | 0.4704 | 0.3622 | 97.803 | 0.0162 | - | - |
| | InFusion | 40m | 0.4943 | 0.3591 | 89.621 | 0.0148 | 203.31 | 0.0601 |
| | InstaInpaint(Ours) | 0.4s | 0.4147 | 0.3130 | 84.535 | 0.0135 | 198.54 | 0.0613 |

Table 2: **Quantitative comparisons with LRM-based methods**. The best and second-best results are highlighted, respectively.

| Category | Method | SPIn-NeRF | | | | LLFF | |
| | | LPIPS↓ | M-LPIPS↓ | FID↓ | KID↓ | C-FID↓ | C-KID↓ |
|---|---|---|---|---|---|---|---|
| None Referenced-Based | SD+LRM | 0.4802 | 0.3645 | 108.43 | 0.0189 | 215.32 | 0.0705 |
| Referenced-Based | MVInpainter+LRM | 0.4122 | 0.3147 | 90.247 | 0.0163 | 217.90 | 0.0758 |
| | InstaInpaint(Ours) | 0.4147 | 0.3130 | 84.535 | 0.0135 | 198.54 | 0.0613 |

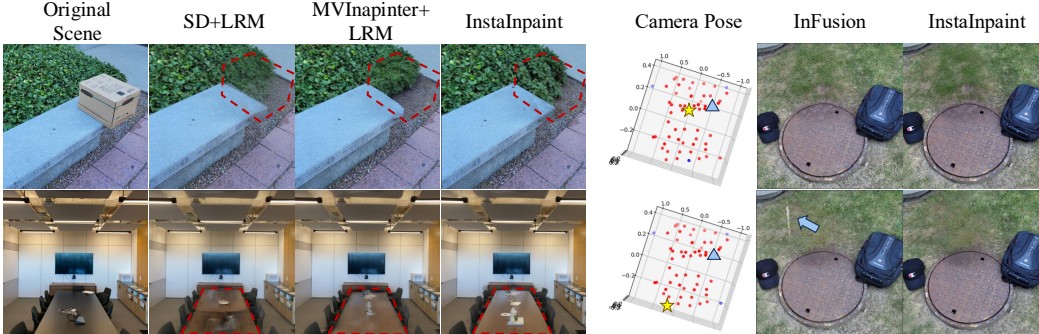

Figure 7: **Qualitative comparisons with LRM-based methods**. InstaInpaint gives cleaner and sharper results with better cross-view consistency.

Figure 8: **Robustness to reference view selection.** The yellow star and blue triangle represent reference views and novel views.

plane. We then compute the convex hull of these projected 2D camera positions. The vertices of this convex hull represent the outermost camera viewpoints. From the vertices of the convex hull, we identify the three points that form a triangle with the largest area. Specifically, this is done by applying the rotating calipers algorithm on the convex hull's endpoints.

**Evaluatied Methods**. We compare against state-of-the-art 3D real-world scene inpainting methods: 1) NeRF based: SPIn-NeRF [30], SPIn-NeRF+SD and MALD-NeRF [32], in which SPIn-NeRF+SD means substituting the inpainting model from LaMA [47] to a diffusion model [48]; 2) 3D-Gaussian based: GScream [35] and InFusion [39]. In addition, we introduce two intuitive baselines using LRM: 1) SD+LRM first inpaints each input view with diffusion inpainter [46] and then uses GS-LRM for 3D reconstruction; 2) MVInpainter+LRM applies a multi-view-consistent 2D inpainting model [10]. We use a state-of-the-art 2D diffusion model [46] to generate the same reference image for fair comparisons with reference-based methods, including GScream, InFusion and MVInpainter+LRM.

**Metrics**. Following prior works [32, 30], we use LPIPS [49], M-LPIPS, FID [50] and KID [51] on SPIn-NeRF dataset and C-FID, C-KID on LLFF dataset. M-LPIPS masks out pixels outside of inpainted masks. FID and KID measure the distributional similarity between two sets of images. Compared with LPIPS, FID/KID is more suitable in 3D inpainting tasks because inpainted areas can have valid content even if it is completely different from the ground-truth testing image. C-FID and C-KID measure visual quality in the inpainting border, since there is no ground-truth value where the foreground object is physically removed for LLFF.

Table 3: **Comparison with reference-based methods using ground-truth image as reference.** InstaInpaint demonstrates better adherence to the reference image on novel views.

| Method | PNSR↑ | M-PNSR↑ | SSIM ↑ | M-SSIM↑ | LPIPS↓ | M-LPIPS↓ |
|---|---|---|---|---|---|---|
| MVInpainter+LRM | 21.182 | 22.010 | 0.5782 | 0.6471 | 0.1741 | 0.2332 |
| Infusion | 19.776 | 20.956 | 0.3550 | 0.04880 | 0.3738 | 0.2596 |
| InstaInpaint(Ours) | 22.799 | 23.881 | 0.6684 | 0.7331 | 0.1684 | 0.1180 |

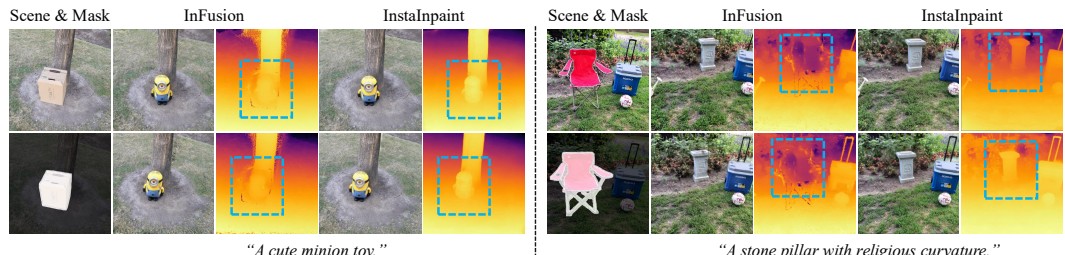

*"A cute minion toy."*    *"A stone pillar with religious curvature."*

Figure 9: **Qualitative comparisons of object insertion ability.** InstaInpaint predicts plausible geometry for inserted objects and seamlessly integrates inpaint regions with reconstruction regions.

## 4.1 Main Results

**Comparison with State-of-the-art Methods.** As shown in Tab. 1 and Fig. 6, InstaInpaint performs favorably against state-of-the-art methods both quantitatively and qualitatively. Compared with optimization-based methods, InstaInpaint reconstructs the scene in a feedforward manner, gaining a $1000\times$ speedup. InstaInpaint also provides a smoother transition at the inpainting border. Please refer to our project page for video results.

**Comparison with LRM-based Methods.** As shown in Tab. 2, InstaInpaint outperforms two LRM-based baselines in FID/KID-related scores. As illustrated in Fig. 7, InstaInpaint produces sharper and geometrically more coherent results than the two proposed LRM-based baselines.

**Using Ground-truth Image as Reference.** FID/KID scores are highly dependent on 2D diffusion inpaint proposal. To better evaluate adherence to the provided reference image for reference-based methods, we provide ground-truth images as reference and evaluate on pixel-level metrics. As shown in Tab. 5, InstaInpaint demonstrates a competitive edge over other reference-based methods.

**Robustness to Reference View Selection.** The selection of the reference image is a crucial factor for reference-based inpainting methods. As shown in Fig. 8, Infusion performs well, given the center of the scene as a reference, but produces noticeable artifacts with a reference image close to the edges of the scene. InstaInpaint performs consistently well in both cases.

**Object Insertion Ability.** Instainpaint can be easily extended to text-driven object insertion tasks by using a text-driven diffusion inpainter to provide 2D reference. In Fig. 9, we show that the baseline method fails to predict the correct geometry and paste the inpainted texture onto the background like a sticker, while InstaInpaint can accurately deduce the geometry of the complete inpainted object and seamlessly merge it into the original scene.

**Multi-region Inpainting.** We show in Fig. 10 that instainpaint can produce a consistent inpainted scene even with multiple disjoint inpainting regions.

## 4.2 Ablation Studies

**Mask Type Selection.** In Tab. 4, we ablate three types of multi-view masks we propose in Section 3.3. The introduction of geometric masks and random image masks mitigates object bias and effectively narrows train-val gaps, yielding better LPIPS, FID and KID scores. Although training with or without object masks produces similar quantitative metrics, we observe that object masks significantly enhance geometric consistency for inserted instances, as shown in Fig. 11. This can be attributed to the object masks' strong cross-view consistency, constraining the model to maintain rigid object structures. Training without object masks results in more deformable geometry of foreground instances (the tilted traffic cone and the twisted minion).

Table 4: **Ablation study on mask types and mask encoding methods.** We underline our default settings in the method section. The best and second-best results are highlighted, respectively.

| Method | | SPIn-NeRF | | | | LLFF | |
|---|---|---|---|---|---|---|
| | | LPIPS↓ | M-LPIPS↓ | FID↓ | KID↓ | C-FID↓ | C-KID↓ |
| Mask Type Selection | w/o object | 0.4132 | 0.3113 | 85.645 | 0.0137 | 198.68 | 0.0603 |
| | w/o random | 0.4185 | 0.3149 | 87.177 | 0.0144 | 200.23 | 0.0634 |
| | w/o geometric | 0.4214 | 0.3170 | 88.565 | 0.0154 | 198.79 | 0.0632 |
| | w/ three types of mask | 0.4147 | 0.3130 | 84.535 | 0.0135 | 198.54 | 0.0613 |
| Mask Encoding Method | only reference view | 0.4247 | 0.3177 | 87.710 | 0.0147 | 199.43 | 0.0618 |
| | all views | 0.4156 | 0.3146 | 86.442 | 0.0140 | 199.56 | 0.0587 |
| | inpaint views | 0.4147 | 0.3130 | 84.535 | 0.0135 | 198.54 | 0.0613 |

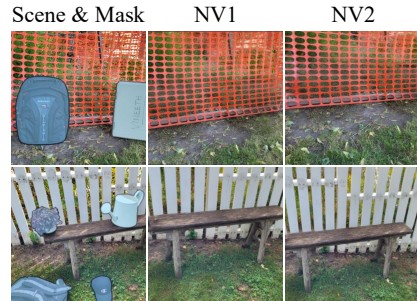

Scene & Mask    NV1    NV2

Figure 10: **Multiple region inpainting.** InstaInpaint supports multiple disjoint inpainting regions.

Figure 11: **Ablation on object masks.** Training without object masks produces deformable geometry across different novel views for inserted objects.

**Mask Encoding Method.** In Tab. 4, we compare three variants of masking encoding options: 1) providing inpaint masks only for the reference view, 2) providing masks for all four views, and 3) providing masks only for three inpaint views. We find that providing masks only for three inpaint views yield the best performance. This can be attributed to the uniform token distribution of reconstruction regions and the targeted inpainting regions, as discussed in Section 3.2.

## 5  Conclusion

**Limitation:** Although InstaInpaint produce high-quality results for static scene inpainting, its performance degrades when handling dynamic scenes with fast-moving objects, the reconstructions become blurry as the model averages conflicting information. Furthermore, if a dynamic object is occluded in the reference view but visible from other context views, the inferred content for the inpainted area may largely contradict the observations from those surrounding views, leading to huge visual inconsistencies. Addressing the challenge of 4D scene (space-time) reconstruction and inpainting for dynamic elements remains an active and complex research area. Future work will investigate integrating InstaInpaint with models specifically designed for dynamic 4D scene reconstruction or incorporating explicit temporal modeling capabilities to mitigate these issues.

We present InstaInpaint, a reference-based feed-forward framework that produces 3D-scene inpainting from a 2D inpainting proposal within 0.4 seconds. By leveraging a self-supervised mask-finetuning strategy, InstaInpaint effectively adapts Large Reconstruction Models for 3D inpainting. InstaInpaint achieves a 1000× speed-up from prior methods while maintaining a state-of-the-art performance across two standard benchmarks and demonstrates strong flexibility for multiple editing applications.

## 6  Acknowledgement

Supported by the Intelligence Advanced Research Projects Activity (IARPA) via Department of Interior/ Interior Business Center (DOI/IBC) contract number 140D0423C0074. The U.S. Government is authorized to reproduce and distribute reprints for Governmental purposes notwithstanding any copyright annotation thereon. Disclaimer: The views and conclusions contained herein are those of the authors and should not be interpreted as necessarily representing the official policies or endorsements, either expressed or implied, of IARPA, DOI/IBC, or the U.S. Government.

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

# Supplementary Material

## A  More Details

**Camera Normalization and Selection.**  For SPIn-NeRF [30] and LLFF [45], we normalize all cameras in a scene into a $[-1,1]^3$ world space. We first calculate the mean of all camera locations and make all poses relative to the mean pose. We then scale all the poses based on the maximum camera deviation. During training, we perform the same normalization procedure on each DL3DV [9] clip (i.e. 15 frames). During validation, we use an intuitive way to select the 4 input views. We choose the camera closest to the mean position for the reference viewpoint. We choose another 3 cameras from the remaining camera positions so that they span a triangle with maximum coverage.

**Oval Mask Generation Startegy.** The random oval masks are generated on the fly during training. For each oval, we first randomly determine its center within the image boundaries. The major and minor axes of the ellipse are sampled with their sizes constrained to be within a certain proportion of the image dimensions to ensure masks are neither too small nor too large. A random angle between 0 and 360 degrees is selected to introduce rotational variability. We randomly sample 1 to 4 ovals and combine them to formulate the final geometric mask.

**Masking Finetuning.** During training, we randomly selects one of three mask types at each iteration: instance masks, geometric masks, or random masks, with sampling probabilities of 25%, 25%, and 50% respectively. We provide ablation for the sampling rate Section B. After choosing the mask type, we randomly apply 1 to 4 masks on to the inpaint views. For random masks, we randomly sample rectangular masks with edge length in range $[\text{size}/6, \text{size}/4]$. For geometric masks, we define elliptical regions with axis lengths determined by the mask count. We sample the two axis from range $[\text{size}/8, \text{size}/6]$ if the mask number is 1 or 2, and $[\text{size}/12, \text{size}/8]$ if the mask number is 3 or 4.

## B  More Ablation Studies

**Multi-region Inpainting Results.** We extend the SPIn-NeRF benchmark to support multi-region inpainting evaluation. Specifically, for each scene in the SPIn-NeRF benchmark, we randomly sample three oval masks on the reference view. We then warp the masks to other views the same way as the Geometric Masks in the paper (with metric depths predicted by LRM trained in the first stage). During Inference, we use the original image from the reference view as reference. We measure pixel-level metrics (PNSR, SSIM, LPIPS) on SPIn-NeRF's testing novel views. In the default training setup, InstaInpaint processes input masks with 1 to 4 masking regions. We find that the number of masking regions combined during training influences the model's performance on multi-region inpainting tasks.

Table 5: **Ablation Study on trainging-time mask numbers.** We underline our default settings in the method section. The best result are highlighted

| Method | PNSR↑ | M-PNSR↑ | SSIM↑ | M-SSIM↑ | LPIPS↓ | M-LPIPS↓ |
|---|---|---|---|---|---|---|
| 1 masking region | 21.463 | 23.019 | 0.6493 | 0.7012 | 0.1729 | 0.1325 |
| 1-4 masking regions | 21.901 | 24.298 | 0.6574 | 0.7253 | 0.1628 | 0.1193 |

**Mask sampling probability.** We provide an ablation study on different mask sampling strategies in Table 6. We can find that the introduction of geometric and random masks effectively improves inpainting performance. The optimal mask sampling distribution is allocating 50% probability to random masks and 25% each to geometric and object masks. This distribution best balance random masks' ability to prevent inconsistent inpainting, geometric masks' preservation of spatial relationships and object masks' high multiview consistency.

## C  More Qualitative Comparisons and Visualizations

Please check our project page for a more straightforward comparison of inpainting consistency. We provide in the video: comparison on SPIn-NeRF and LLFF with previous 3D inpainting methods [32, 35, 39], comparison on object insertion ability and more text-guided 3D inpainting results.

Table 6: **Ablation study on mask sampling probability.** We underline our default settings in the method section. The best and second-best results are highlighted, respectively.

| Object Mask | Geometric Mask | Random Mask | SPIn-NeRF | | | | LLFF | |
|---|---|---|---|---|---|---|---|---|
| | | | LPIPS↓ | M-LPIPS↓ | FID↓ | KID↓ | C-FID↓ | C-KID↓ |
| 33% | 33% | 33% | 0.4289 | 0.3532 | 86.432 | 0.0139 | 199.25 | 0.0628 |
| 50% | 25% | 25% | 0.4326 | 0.3678 | 86.973 | 0.0142 | 199.89 | 0.0623 |
| 25% | 50% | 25% | 0.4191 | 0.3484 | 85.385 | 0.0132 | 198.94 | 0.0611 |
| 25% | 25% | 50% | 0.4147 | 0.3130 | 84.535 | 0.0135 | 198.54 | 0.0613 |

# D  Boarder Impact

Our work bridges 3D reconstruction and inpainting by adapting Large Reconstruction Models for real-time 3D completion. This advancement enables two key opportunities:

First, when combined with 2D generative models, our system allows instant creation of 3D-consistent assets from text prompts - particularly valuable for VR/AR applications where users can interactively modify 3D environments.

Second, the technical approach demonstrates how reconstruction-focused models can be repurposed for generative tasks, suggesting a brand new research direction.

Admittedly, like any technology, our method could potentially be misused to generate manipulated 3D reconstructions or factually inaccurate renderings. We recognize the need for safeguards and mechanisms to attribute AI-generated assets.

