# OpenReview forum: "InstaInpaint: Instant 3D-Scene Inpainting with Masked Large Reconstruction Model"
_NeurIPS.cc/2025/Conference — NeurIPS 2025 poster_

### Official Review · Reviewer_6csA · 2025-06-02

**Clarity:** 4
**Significance:** 3
**Originality:** 2
**Rating:** 4
**Confidence:** 5

**Summary:**

This paper propose InstaInpaint, a novel inpainting framework that restores missing regions of 3D scenes based on multi-view images. The ViT-based large-scale reconstruction model is trained in two stages, and the second stage enhances the inpainting ability through mask-based fine-tuning. Various mask generation strategies (object, geometric, and random masks) are introduced to achieve both data efficiency and generalization. Experiments show that InstaInpaint provides more consistent and natural 3D inpainting results than existing methods.

**Questions:**

The proposed method shows promising results in real-world sparse-view scenarios. However, how does InstaInpaint perform when the number of input views increases significantly? Can the method generalize or scale to dense-view reconstruction tasks?

While the proposed training strategy is effective, the model architecture remains largely similar to MaskedLRM. Could the authors clarify whether any architectural innovations were introduced, and if so, how they specifically contribute to performance improvements?

Recent work such as SimVS [1] addresses multi-view inconsistencies explicitly through simulation and robustness training. How does InstaInpaint differ conceptually or empirically from SimVS? Are there any comparative experiments to justify its advantages in handling inconsistent inputs?

The paper lacks direct comparisons with other multi-view 2D inpainting-based methods, such as InpaintNeRF360, SPIn-NeRF, or GScream. Can the authors provide additional quantitative results or ablation studies to better position their work among existing techniques?

Traditional metrics like PSNR and SSIM are not reported or discussed. Could the authors clarify why these metrics were omitted, and whether the proposed method was evaluated under those metrics to ensure fair comparison with prior inpainting works?

**Ethical Concerns:**

["NO or VERY MINOR ethics concerns only"]

**Final Justification:**

The rebuttal has addressed all the concerns I raised. The clarifications regarding the sparse-view setting, architectural contributions, comparison with SimVS, quantitative evaluations, and the use of PSNR/SSIM are convincing and well-reasoned.

**Limitations:**

Yes

**Quality:**

2

**Strengths And Weaknesses:**

Strengths
1. To overcome the limitations of existing LRM-based models, this approach proposes a self-supervised masked fine-tuning strategy centered on reference images to enable effective 3D inpainting in real-world scenes.
2. By combining object masks, geometric masks, and random masks and using them as training masks, it simultaneously improves the generalization performance and robustness of the model.
3. By utilizing multi-view masks and photometric loss during the learning process, it induces the restored area to be naturally reconstructed from various viewpoints, thereby solving the unnatural boundary and floating object problems of existing methods.

Weaknesses
1. InstaInpaint is tailored for sparse-view settings and does not scale well to scenarios with a large number of input images, limiting its applicability in dense multi-view reconstruction tasks.

2. InstaInpaint adopts a vision transformer-based architecture similar to MaskedLRM, but lacks fundamental architectural differences and no clear structural improvements.

3. SimVS (CVPR 2025) simulates multi-view inconsistency to enhance the robustness of learning, while InstaInpaint does not directly address this kind of challenging setting and lacks comparative experiments or differentiation strategies with SimVS.

4. Quantitative comparison with existing multi-view 2D inpainting-based NeRF/3DGS studies is limited, and the persuasiveness of the argument needs to be strengthened through head-to-head performance comparison with competing techniques.

5. The reasons for not using or ignoring traditional performance metrics such as PSNR and SSIM, which are widely used in existing inpainting literature, are not clearly described. This makes it difficult to compare the quality of the proposed technique and may raise questions about its objectivity.

[1]SimVS: Simulating World Inconsistencies for Robust View Synthesis. CVPR 2025

---

> ### Author Rebuttal · Authors · 2025-07-31
>
> We thank reviewer 6csA for the valuable suggestions.
>
> **W1/P1:** Limitation to sparse view settings
>
> As acknowledged in our limitation section, InstaInpaint is indeed limited to a sparse-view setting and the performance degrades when the input view number increases.
>
> However, the sparse-view setting also represents a feature for many real-world downstream applications where capturing dense multi-view data is impractical (e.g., mobile capture) or redundant (e.g., benchmarks such as SPIn-NeRF where the scenes are overly densely captured).
>
> Regarding scaling to a larger number of input views and dense-view reconstruction tasks, one potential method to scale InstaInpaint to a larger number of input views is by applying the masking finetuning pipeline on dense-view feed-forward reconstruction model such as Long-LRM [1].
>
> **W2/P2:** Lack of architectural improvements
>
> We acknowledge that InstaInpaint does not introduce novel network architecture to GS-LRM model. However, our innovation centers on a flexible masking fine-tuning pipeline and its specific mask designs that effectively adapt reconstruction models to the 3D inpainting task. As a data-driven method, we believe that this pipeline's ability to leverage and guide existing powerful models is a significant and equally important contribution to the field.
>
> Furthermore, while both InstaInpaint and MaskedLRM share architectural similarities with GS-LRM, we differ significantly in 3D representation and target application. MaskedLRM targets object-centric mesh editing and represents the 3D object in triplane-NeRF. InstaInpaint targets real-world scene editing, utilizing 3D Gaussian Splatting as its scene representation. The complexity of real-world scenes forces us to exploit more sophisticated masking strategies to adapt real-world multi-view datasets to 3D inpainting tasks.
>
> **W3/P3:** Comparison with SimVS.
>
> Thank you for bringing SimVS [2] to our attention and for asking about the differentiation.
>
> The difference between SimVS and InstaInpaint highlights the benefit and the design philosophy of InstaInpaint. SimVS handles inherently 3D inconsistent multi-view images. On the other hand, InstaInpaint aims to lift 2D-generated content to 3D using 3D consistent context from the non-inpainting area. Our design naturally avoids resolving the challenging 3D inconsistency arising from multi-view inpainting models. We also show the benefit of our approach by comparing with "MVInpainter+LRM" (in Table 3), where the moderate 3D inconsistency of this approach leads to a more blurry appearance and less favorable performance.
>
> Unfortunately, we are unable to compare to SimVS, as the model is not open-sourced. However, the "MVInpainter+LRM" baseline is highly relevant to the role of SimVS in the 3D inpainting problem: using a learned cross-view consistent diffusion model as reconstruction target, then performing feed-forward 3D reconstruction of the inpainted object. MVInpainter is our best effort in finding an open-sourced cross-view consistent generative model. However, MVInpainter produces very inconsistent results in high-frequency areas, leading to sub-optimal results compared to InstaInpaint, as is shown in Figure 3, Figure 7 and Table 2. This suggests a fundamental challenge in achieving perfect multi-view consistency solely through 2D generation for 3D reconstruction.
>
> **W4/P4:** Direct comparision with multi-view 2D inpainting-based methods.
>
> We would like to clarify that **we have provided direct quantitative results with SPIn-NeRF, MALD-NeRF, GScream and InFusion in Table 1.** We add comparison with InpaintNeRF360 in the table below.
>
> | Method | LPIPS(SPIn-NeRF) | MLPIPS(SPIn-NeRF) | FID(SPIn-NeRF) | KID(SPIn-NeRF) | C-FID(LLFF) | C-KID(LLFF) |
> |---|---|---|---|---|---|---|
> | InpaintNeRF360 | 0.4562 | 0.3417 | 117.43 | 0.0216 | 229.85 | 0.0712 |
> | InstaInpaint | **0.4147** | **0.3130** | **84.535** | **0.0135** | **198.54** | **0.0613** |
>
> **W5/P5:** Experiments with PSNR and SSIM metric
>
> First, we would like to clarify that **we do report PSNR, SSIM and LPIPS for both InstaInpaint and other reference-based 3D inpainting works (Including Infusion and MVInpainter+LRM) in Table 3.** We further provide a comparison with GScream in the table below.
>
> | Method | PNSR | MPNSR | SSIM | MSSIM | LPIPS | MLPIPS |
> |---|---|---|---|---|---|---|
> | GScream | 20.425 | 21.323 | 0.5696 | 0.6234 | 0.2341 | 0.2146 |
> | InstaInpaint | **22.799** | **23.881** | **0.6684** | **0.7331**  | **0.1684**  | **0.1180** |
>
> Below, we will explain why PNSR and SSIM can only be used to compare reference-based inpainting methods. PNSR and SSIM are low-level metrics that directly measure the resemblance between pixels.
>
> **For non-reference inpainting methods, the content within the inpainted area is non-deterministic.** This means there can be multiple perceptually plausible and equally "correct" ways to fill a missing region, even if their pixel values differ significantly from the specific ground truth. In such scenarios, a low PSNR or SSIM score does not necessarily indicate poor perceptual quality, as the generated result might be reasonable despite differing from the exact ground truth pixels. For these methods, higher-level perceptual metrics (such as LPIPS and FID, which we include in Table 1 of the manuscript) are often more appropriate. We note that existing non-reference inpainting methods, such as SPIn-NeRF and MALD-NeRF, do not report PNSR and SSIM.
>
> In the context of reference-based inpainting methods, the "inpainting" task is constrained by a reference. By providing the ground truth texture as a reference to these models, we effectively make the target content in the inpainted region more deterministic. For our evaluations in Table 3, we use the ground-truth image as this explicit reference. By providing this ground-truth content, the target for the inpainted region becomes deterministic, enabling PSNR and SSIM to accurately measure how closely the method reconstructs the specific ground-truth pixels at novel views.
>
> [1] Long-LRM: Long-sequence Large Reconstruction Model for Wide-coverage Gaussian Splats. Chen, et al. ICCV 2025.
>
> [2] SimVS: Simulating World Inconsistencies for Robust View Synthesis. Trevithick, et al. CVPR 2025.

---

> > ### Author Response · Authors · 2025-08-04
> >
> > Dear Reviewer,
> >
> > We sincerely appreciate your time and effort in reviewing our manuscript and offering valuable suggestions.
> >
> > As we near the midpoint of our discussion period, we would like to confirm whether our responses have effectively addressed your concerns. If you require further clarification or have any additional concerns, please do not hesitate to contact us. We are happy to provide any further information you may need.
> >
> > Thank you.

---

> > > ### Comment · Area_Chair_cwo2 · 2025-08-05
> > >
> > > Dear Reviewer 6csA,
> > >
> > > Based on the original reviews for this paper, your rating was the lowest among the four reviews, so we would definitely like to hear from you during the discussion phase. Could you read the rebuttal from the authors? Does it address your questions and concerns?
> > >
> > > Best,
> > > AC

---

> > ### Comment · Reviewer_6csA · 2025-08-06
> >
> > Thank you for the authors for the detailed and thoughtful responses to the comments.
> >
> > The rebuttal has addressed all the concerns I raised. The clarifications regarding the sparse-view setting, architectural contributions, comparison with SimVS, quantitative evaluations, and the use of PSNR/SSIM are convincing and well-reasoned.

---

> > > ### Author Response · Authors · 2025-08-06
> > >
> > > Dear reviewer 6csA,
> > >
> > > Thank you for all your feedback. We will include the discussions in the revised paper. Given that all concerns have now been resolved, we would be most grateful if you could consider raising your rating accordingly.
> > >
> > > Thank you

---

### Official Review · Reviewer_JFCW · 2025-06-22

**Clarity:** 3
**Significance:** 2
**Originality:** 3
**Rating:** 5
**Confidence:** 4

**Summary:**

This paper proposes InstaInpaint, a novel 3D scene inpainting framework based on a feedforward large reconstruction model (LRM). The authors introduce a self-supervised masked fine-tuning strategy and demonstrate strong performance in both speed and quality on standard 3D inpainting benchmarks. The method achieves a remarkable 1000× speed-up over prior optimization-based approaches and supports multiple tasks including object removal, insertion, and multi-region inpainting.

**Questions:**

(1) The abstract emphasizes multi-region inpainting as a key feature, yet the main body only briefly touches on this in Figure 10. Please provide detailed technical explanations (e.g., mask handling, training design) and more extensive quantitative and qualitative evaluations to validate this claim.
(2) Given that InstaInpaint is built on GS-LRM, the paper would benefit from clearer justification or discussion of any architectural innovations beyond standard LRM usage.
(3) The discussion of limitations in Section 5 is very brief. Could the authors provide additional examples or figures that illustrate common failure modes of InstaInpaint (e.g., dynamic scenes, inconsistent masks), and how future work might address these?

**Ethical Concerns:**

["NO or VERY MINOR ethics concerns only"]

**Final Justification:**

The rebuttal and discussions with the authors have addressed my concerns. Thus I would support on accepting it.

**Limitations:**

Yes.

**Paper Formatting Concerns:**

None.

**Quality:**

2

**Strengths And Weaknesses:**

Strengths:
(1) The proposed framework is novel in applying a feedforward LRM to the inpainting task, achieving a substantial 1000× speed-up over traditional optimization-based methods.
(2) The training methodology is effective, utilizing three distinct mask generation strategies (object masks, geometric masks, and random image masks) to construct diverse and scalable self-supervised training data.
(3) The experimental section is comprehensive. The proposed approach achieves state-of-the-art performance on standard 3D inpainting benchmarks, both in terms of quality and speed.
Weaknesses:
(1) Although the abstract claims support for multi-region inpainting, the main text lacks detailed technical descriptions and corresponding experimental results that explicitly evaluate this capability.
(2) The framework heavily relies on GS-LRM architecture; the paper introduces limited innovation in network design, and the application of LRM appears relatively straightforward without significant architectural modifications.
(3) The claimed 1000× acceleration is largely attributed to the use of LRM, rather than innovations introduced by the authors in this paper.

---

> ### Author Rebuttal · Authors · 2025-07-31
>
> We thank reviewer JFCW for the valuable suggestions.
>
> **W1/P1:** More details and experiments on multi-region inpainting.
>
> We mention InstaInpaint's multi-region inpainting to emphasize its flexibility and generalizability that various tasks can be supported without much specific design. However, we find that the number of training masks is relevant to model performance on multi-region inpainting. In our current training setup, the model processes input masks with 1 to 4 masking regions. We find that the number of masking regions combined during training influences the model's performance on multi-region inpainting tasks.
>
> We add an ablation study on the number of masking regions during the training stage. We extend the SPIn-NeRF benchmark to support multi-region inpainting evaluation. Specifically, for each scene in the SPIn-NeRF benchmark, we randomly sample three oval masks on the reference view. We then warp the masks to other views the same way as the Geometric Masks in the paper (with metric depths predicted by LRM trained in the first stage). During Inference, we use the original image from the reference view as reference. We measure pixel-level metrics (PNSR, SSIM, LPIPS) on SPIn-NeRF's testing novel views. The results are presented in the table below:
>
> | Method | PNSR | MPNSR | SSIM | MSSIM | LPIPS | MLPIPS |
> |---|---|---|---|---|---|---|
> | 1 masking region | 21.463 | 23.019 | 0.6493 | 0.7012 | 0.1729 | 0.1325 |
> | 1-4 masking regions | **21.901** | **24.298** | **0.6574** | **0.7253**  | **0.1628**  | **0.1193** |
>
> **W2/P2:** Architectural innovations.
>
> We acknowledge that InstaInpaint does not introduce a novel network architecture to GS-LRM [1] model. Our innovation centers on a flexible masking fine-tuning pipeline. The pipeline and its specific mask designs effectively adapt reconstruction models to the 3D inpainting task. **As a data-driven method, this pipeline's ability to leverage and guide existing powerful models is a significant and equally important contribution to the field.**
>
> **W3:** Acceleration stems from LRM but not our designs
>
> It is true that the 1000x acceleration is indeed powered by the underlying Large Reconstruction Model (LRM). But LRMs are not natively equipped for 3D inpainting. The main innovation is the masking framework designed to overcome the challenges involved in adapting powerful data-driven reconstruction models to the specialized domain of 3D inpainting, including a lack of inpainting dataset, object bias, etc. **We are the first to adapt LRM to 3D inpainting field, unlocking their speed for this complex task, which is a problem not addressed by LRMs alone.**
>
> **P3:** More discussions on limitations.
>
> While we are unable to provide visual examples in this rebuttal, we can elaborate on how dynamic scenes affect InstaInpaint. In dynamic regions, the reconstructions become blurry as the model averages conflicting information. Furthermore, if a dynamic object is occluded in the reference view but visible from other context views, the inferred content for the inpainted area may largely contradict the observations from those surrounding views, leading to huge visual inconsistencies. Addressing the challenge of 4D scene (space-time) reconstruction and inpainting for dynamic elements remains an active and complex research area. Future work will investigate integrating InstaInpaint with models specifically designed for dynamic 4D scene reconstruction or incorporating explicit temporal modeling capabilities to mitigate these issues.
>
> We would like to clarify that InstaInpaint is designed to handle mask inconsistencies. The inconsistent random image masks help build robustness to slight inconsistencies introduced by inconsistent maskings.
>
> [1] GS-LRM: Large Reconstruction Model for 3D Gaussian Splatting. Zhang, et al. ECCV 2024.

---

> > ### Comment · Reviewer_JFCW · 2025-08-05
> >
> > The rebuttal has addressed most of my concerns. The revision should include the points mentioned in the rebuttal.

---

> > > ### Author Response · Authors · 2025-08-05
> > >
> > > Thank you for the comments! We will refine our paper accordingly.

---

> ### Author Response · Authors · 2025-08-04
>
> Dear Reviewer,
>
> We sincerely appreciate your time and effort in reviewing our manuscript and offering valuable suggestions.
>
> As we near the midpoint of our discussion period, we would like to confirm whether our responses have effectively addressed your concerns. If you require further clarification or have any additional concerns, please do not hesitate to contact us. We are happy to provide any further information you may need.
>
> Thank you.

---

### Official Review · Reviewer_nJGu · 2025-06-30

**Clarity:** 4
**Significance:** 4
**Originality:** 4
**Rating:** 5
**Confidence:** 4

**Summary:**

This paper presents a 3D inpainting method based on large reconstruction models (LRMs). It introduces a specifically designed LRM pipeline tailored for the multi-view inpainting task. The approach involves fine-tuning an LRM conditioned on one inpainted reference view and three additional masked views to generate Gaussians representing the full scene, including occluded (inpainted) regions. The training of such a model is non-trivial, as it requires multi-view masked image data. To address this, the authors leverage video segmentation priors and geometric priors to generate both object-level and scene-level masks, along with additional random masks. With this fine-tuning strategy, the method achieves effective and efficient 3D inpainting.

**Questions:**

Did the authors try any per-scene optimization, and how does it compare to direct prediction from the LRM? From my understanding, per-scene optimization should lead to improved results in practical scenarios, especially since supervision is available for the unmasked regions.

How would author consider to scale this approach to 360-degree scenes?

**Ethical Concerns:**

["NO or VERY MINOR ethics concerns only"]

**Final Justification:**

The rebuttal thoroughly addresses all of my earlier concerns. The authors provide a clear and detailed explanation of their input view selection strategy, commit to adding missing citations to relevant prior works, and outline feasible extensions for scaling beyond sparse-view settings. They also present new per-scene optimization results showing consistent quality gains with minimal overhead, reinforcing the method’s practicality. With these clarifications and additional evidence, I now find the contribution to be solid, well-motivated, and sufficiently validated, and I am raising my score from borderline accept to accept.

**Limitations:**

Yes

**Quality:**

4

**Strengths And Weaknesses:**

Strengths:

1.	The method effectively addresses the challenge of inconsistency in inpainting using large reconstruction models (LRMs).
2.	The authors propose multiple variants of mask generation strategies to compensate for the lack of multi-view masked data, and these approaches are self-supervised.
3.	Training an LRM for this task is non-trivial; the pipeline appears to involve significant engineering effort and rigorous sanity checks.
4.	The results are promising in terms of both quality and efficiency, and the evaluation is thorough and complete.


Weaknesses:

1.	As the authors acknowledge, the method relies on a fixed number of input images, which limits its scalability to 360 scenes.
2.	The paper does not clearly explain how the reference view is selected and inpainted, nor how the three target views are chosen during the inpainting pipeline described in the experiments.
3.	The paper omits relevant citations and credit to prior works that use similar reference-based inpainting strategies, such as “In-N-Out: Lifting 2D Diffusion Prior for 3D Object Removal via Tuning-Free Latents Alignment” (NeurIPS 2024) and “Imfine: 3D Inpainting via Geometry-Guided Multi-View Refinement” (CVPR 2025).

---

> ### Author Rebuttal · Authors · 2025-07-31
>
> We thank reviewer nJGu for the valuable suggestions.
>
> **W1/P2:** Scalability to 360 scenes.
>
> We acknowledge that InstaInpaint is primarily designed for sparse-view inpainting and reconstruction, and full 360-degree scene reconstruction is indeed beyond its scope. But we have several promising solutions for extending our work to the challenge.
> 1. **Finetuning on dense-view feed-forward reconstruction model (e.g. Long-LRM[1]).** InstaInpaint's mask finetuning pipeline is very flexible and is suitable for feed-forward reconstruction models other than GS-LRM[2].
> 2. **A two-stage 360-degree scene inpainting method.** First, we can optimize a "partial" 3D-Gaussian scene with the inpainting area masked out. The inpainting area is left as a hole to be completed. Then we sample a sparse set of view images as inputs to InstaInpaint. Since Gaussian blobs predicted by InstaInpaint are strictly registered to the camera, we can store the predicted Gaussian blobs from the inpaint area at the original scene scale with a simple affine transformation and merge them into the optimized partial Gaussian Scene. To further enhance quality, especially at the merging boundaries, the combined Gaussian scene could undergo a few additional optimization steps.
>
> **W2:** Details on input views selection in the experiments.
>
> We select the 4 input views to ensure **1) reference view is posed close to the center and 2) the three other context views cover an area as large as possible.**
> 1. **For reference view,** we calculate the mean of all training views (60 for SPIn-NeRF) and select the view whose position is closest to the mean as the reference view. The reference view is inpainted with SOTA 2D-diffusion inpainting models[3].
> 2. **For the three other views**, we aim to choose three views that cover as wide range as possible. We first regress a plane from all training views. Then we project all training view positions onto the plane. We then compute the convex hull of these projected 2D camera positions. The vertices of this convex hull represent the outermost camera viewpoints. From the vertices of the convex hull, we identify the three points that form a triangle with the largest area. Specifically, this is done by applying the rotating calipers algorithm on the convex hull's endpoints.
>
> **W3:** Missing citations.
>
> We will cite and discuss the suggested references in the revised paper.
>
> **P1**: Per-scene Optimization Results
>
> Yes, per-scene optimization does improve reconstruction quality compared with direct prediction from LRM. We experiment to test this.
>
> | Method | PNSR | MPNSR | SSIM | MSSIM | LPIPS | MLPIPS |
> |---|---|---|---|---|---|---|
> | InstaInpaint | 22.799 | 23.881 | 0.6684 | 0.7331  | 0.1684  | 0.1180 |
> | InstaInpaint+post-optimization | **23.252** | **24.452** | **0.6823** | **0.7429** | **0.1592** | **0.1073** |
>
> This experiment is conducted following the setting mentioned in the paper from L241-L244. We optimize the LRM-predicted Gaussians on the 4 input views for 100 steps, which takes roughly 5 seconds. For reference view, we compute photometric loss on the entire image and all Gaussian parameters as learnable. For non-reference views, we compute photometric loss on the unmasked region and set Gaussians within the inpainted region as unlearnable.
>
> [1] Long-LRM: Long-sequence Large Reconstruction Model for Wide-coverage Gaussian Splats. Chen, et al. ICCV 2025.
>
> [2] GS-LRM: Large Reconstruction Model for 3D Gaussian Splatting. Zhang, et al. ECCV 2024.
>
> [3] FLUX. Black Forest Lab.

---

> > ### Author Response · Authors · 2025-08-04
> >
> > Dear Reviewer,
> >
> > We sincerely appreciate your time and effort in reviewing our manuscript and offering valuable suggestions.
> >
> > As we near the midpoint of our discussion period, we would like to confirm whether our responses have effectively addressed your concerns. If you require further clarification or have any additional concerns, please do not hesitate to contact us. We are happy to provide any further information you may need.
> >
> > Thank you.

---

> > ### Comment · Reviewer_nJGu · 2025-08-07
> >
> > Thank you authors for the detailed and thoughtful rebuttal. I believe my concerns have been addressed. I’ll raise my score accordingly.

---

### Official Review · Reviewer_hGee · 2025-07-03

**Clarity:** 3
**Significance:** 3
**Originality:** 3
**Rating:** 5
**Confidence:** 4

**Summary:**

This paper presents a single-stage feed-forward method for 3D reconstruction and editing, which supports both 3D inpainting and object replacement.

The authors introduce a self-supervised masked fine-tuning scheme to train the model. In the fine-tuning stage, masked, incomplete multi-view images are used as inputs, while original images from other views serve as supervision for generating the 3D scenes.
To generate the masks for the fine-tuning process, InstaInpaint employs three different types: Object Mask, Geometric Mask, and Random Image Mask.

Experimental results demonstrate that InstaInpaint can perform view-consistent 3D editing in under 1 second while maintaining high-quality inpainting performance. The ablation studies further confirm the effectiveness of the proposed mask types used during the fine-tuning stage.

**Questions:**

1. Could this method be applied in autonomous driving scenarios? Adding objects into street scenes would be valuable for simulations in driving environments. Demonstrating successful applications in such scenarios would further highlight the impact and versatility of this method.

2. Was the inpainting mask generation process inspired by the training stage of image inpainting models? If so, it would be helpful to reference related works in this area.


My concerns primarily arise from some unclear descriptions. I would be happy to raise my score if the authors could clarify the points I have mentioned.

**Ethical Concerns:**

["NO or VERY MINOR ethics concerns only"]

**Final Justification:**

Most of my concerns have been well addressed in the detailed replies, and I appreciate the updated, clear explanations of the different tokens—these have greatly enhanced my understanding of the processing procedure. Considering all the reviews and responses, I've raised my score to accept.

**Limitations:**

Yes

**Paper Formatting Concerns:**

No paper formatting concerns.

**Quality:**

3

**Strengths And Weaknesses:**

### Strengths:
1. The paper is well-written and easy to follow, with key technical details provided that help in understanding the proposed method.
2. The proposed method achieves better 3D inpainting results than the baseline methods, with significantly faster inference speeds.
3. The use of video segmentation methods and GS-LRM to generate data for 3D inpainting training could inspire future research.


### Weakness:
1. Typo: "InstaInapint" in Figure 1 (should be "InstaInpaint").
2. Including NeRFiller [A] and O2-Recon [B] in the related works section on 3D scene inpainting would strengthen this part. These methods use image grids and mask reprojection techniques to enhance the multi-view consistency of inpainted results.
3. The explanation in lines L142-L149 is unclear. Since the authors state that there are no masked regions in the reference image in L142, it’s hard to understand what (2) in L145 means.
4. It is unclear how the method selects one or more ovals, as described in L182. A more detailed explanation would be helpful.
5. The motivation behind using the "Random Image Mask" in the fine-tuning stage needs further clarification. Specifically, why does it improve the model’s robustness?


### References:
A. Weber, Ethan, et al. "Nerfiller: Completing scenes via generative 3d inpainting." Proceedings of the IEEE/CVF Conference on Computer Vision and Pattern Recognition. 2024.
B. Hu, Yubin, et al. "O^ 2-recon: completing 3D reconstruction of occluded objects in the scene with a pre-trained 2D diffusion model." Proceedings of the AAAI Conference on Artificial Intelligence. Vol. 38. No. 3. 2024.

---

> ### Author Rebuttal · Authors · 2025-07-31
>
> We thank reviewer hGee for the valuable suggestions.
>
> **W1:** Typo in Figure 1.
>
> We will fix the typo in the revised paper.
>
> **W2:** Missing Related Works.
>
> We will cite and discuss the suggested references in the revised paper.
>
> **W3:** Clarification on L142-L145.
>
> We revise and expand L142-L145 for better clarity.
>
> We do not provide the inpaint region mask directly to the reference view. This means that when the reference image is fed into our pipeline, the binary mask image indicating the inpaint region is not given as an additional input alongside the reference RGB image itself, nor is the reference image's content explicitly altered (e.g., grayed out) in the masked region.
>
> This design is inspired by a key observation: the inpainted region and the reconstruction regions should maintain a similar data distribution during training. To clarify the model's perception of different areas, we conceptualize three kinds of tokens within our pipeline:
>
> **(1) Unmasked regions with intact texture:** These areas are fully visible and consistent across both reference and supervision views, providing clear multi-view stereo information vital for accurate 3D reconstruction.
>
> **(2) "Masked regions" in the reference view with original texture:** This refers to the conceptual area within the reference view that corresponds to the region proposed for inpainting. The model "sees" the original, unaltered texture in this region. However, since other non-reference views do have these corresponding regions explicitly masked out, the model is implicitly tasked with inferring the geometry and fine details of this area within the reference view based solely on its surrounding unmasked context. **This unique setup is vital for maintaining a similar data distribution between token types (1) and (2), enabling the model to learn to faithfully reconstruct both areas equally well, whether the mask is explicitly presented or not.**
>
> **(3) Masked regions in non-reference views filled with gray pixels**: These regions in non-reference views are explicitly altered (grayed out) with a provided mask and are provided with an explicit binary mask. This clear alteration forces the model to learn to ignore these areas or infer their content from other available views.
>
> We aim for the model to faithfully reconstruct (1) using multi-view stereo vision, to accurately speculate the geometry of (2) based on neighboring regions in the reference view, and to mitigate negative effects from (3). This approach ensures a sufficient distinction between (2) and (3) to prevent feature contamination while promoting consistent learning across visible and conceptually "masked" areas.
>
> **W4:** Generation details on random oval masks for geometric masks.
>
> The random oval masks are generated on the fly during training. For each oval, we first randomly determine its center within the image boundaries. The major and minor axes of the ellipse are sampled with their sizes constrained to be within a certain proportion of the image dimensions to ensure masks are neither too small nor too large. A random angle between 0 and 360 degrees is selected to introduce rotational variability. We randomly sample 1 to 4 ovals and combine them to formulate the final geometric mask.
>
> **W5:** Clarification on the motivation of random image masks.
>
> We will include more design motivations in the updated paper. Here are two main motivations:
> - **Test-time 3D inconsistency.** The testing benchmarks have their own annotation strategies different from our training strategy. The human annotation errors and mask dilation strategy often create substantial leftover masks bleeding through the actual object boundary, leading to 3D inconsistent masks in the background. Our random masks provide distinctive learning signals aside from the other two 3D consistent masking strategies.
> - **Data diversity.** Similar to we address the object bias of object masks using randomized geometric masks. The geometric mask still follows certain biased data distribution and potentially introduces unattended biases. Our random masks provide challenging 3D inconsistent masks for the model to resolve the conflicts through finding 3D correspondences across images. We believe this strategy increases the data diversity and augments important learning tasks for the model to learn a more robust and generalizable representation.
>
> **P1:** Potential application in autonomous driving.
>
> InstaInpaint could be used for editing static elements in street scenes. For example, one could add or remove fixed objects like traffic signs, streetlights, or parked vehicles. Our method, trained on static scene data, can seamlessly integrate or remove these stationary items while maintaining 3D consistency.
>
> However, applying InstaInpaint to dynamic road users (e.g., moving cars or pedestrians) would be challenging. InstaInpaint is trained for static 3D scene reconstruction and inpainting. Adapting our method for dynamic scene editing is an exciting area for future work.
>
> **P2:** Related work from image editing area.
>
> Masking training indeed is a common technique in image inpainting area, such as RePaint [1] and LaMa [2]. We will add representative works in the related work.
>
>
> [1] RePaint: Inpainting using Denoising Diffusion Probabilistic Models. Lugmayr, et al. CVPR 2022
>
> [2] Resolution-robust Large Mask Inpainting with Fourier Convolutions. Suvorov, et al. WACV 2022

---

> ### Author Response · Authors · 2025-08-04
>
> Dear Reviewer,
>
> We sincerely appreciate your time and effort in reviewing our manuscript and offering valuable suggestions.
>
> As we near the midpoint of our discussion period, we would like to confirm whether our responses have effectively addressed your concerns. If you require further clarification or have any additional concerns, please do not hesitate to contact us. We are happy to provide any further information you may need.
>
> Thank you.

---

> > ### Comment · Reviewer_hGee · 2025-08-05
> >
> > Thank you very much for the authors’ thoughtful responses. Most of my concerns have been well addressed in the detailed replies, and I appreciate the updated, clear explanations of the different tokens—these have greatly enhanced my understanding of the processing procedure. I will consider raising my score following the reviewer discussion phase.

---

> > > ### Author Response · Authors · 2025-08-05
> > >
> > > Thanks for your comments. We are delighted to hear that our responses have addressed your concerns.

---

### Author Response · Authors · 2025-08-02

Dear reviewers,

We would like to verify whether our responses have fully addressed all the concerns in your reviews. If any issue remains that requires additional clarification or modification, please let us know at your earliest convenience. This will enable us to address them promptly.

We appreciate your time and effort in enhancing the quality of our manuscript.

Thank you.

---

> ### Comment · Area_Chair_cwo2 · 2025-08-05
>
> Dear reviewers,
>
> The authors have responded to the original reviews. Could you read the rebuttal and share your thoughts? Does it address your original concerns? Are there any remaining questions for the authors? We still haven't heard from anyone for this paper and we would like to hear from everyone before the end of the discussion phase.
>
> Best,
> AC

---

### Note · Authors · 2025-08-15

We sincerely appreciate the time and effort all reviewers have dedicated to evaluating our manuscript. Their invaluable suggestions have significantly contributed to improving the quality of our work.

During the rebuttal stage, we are pleased to note that the reviewers have confirmed that all of their concerns have been adequately addressed, with no further issues raised. We are grateful for their constructive feedback and the opportunity to refine our manuscript accordingly.

---

### Decision · Program_Chairs · 2025-09-17

**Decision:**

Accept (poster)

**Comment:**

This paper presents a feedforward approach that allows for inpainting and editing of a 3D scene, by leveraging a Large Reconstruction Model (LRM). Initially, the paper received one Reject and three Borderline Accept ratings. The reviewers appreciated the thorough evaluation, the strong results and the challenge in training an LRM for this task. However, there were some remaining concerns, e.g., related to perfomance of the approach when using a large number of input images, the minimal changes to the existing model architecture, missing related work and some unclear design choices. The authors submitted a rebuttal which helped increase the ratings of all reviewers. Eventually, the paper received one Borderline Accept and three Accept ratings. Given the unanimous acceptance recommendation from four knowledgable reviewers, there is no basis to overturn reviews. The AC recommends acceptance. Authors should still consider any additional comments or feedback from the reviewers while preparing their final version and of course update the manuscript to include any additional promised changes, analysis, results and/or discussion they provided in their rebuttal.